

# Improved records of glacier flow instabilities using customized NASA autoRIFT applied to PlanetScope imagery

Jukes Liu[1], Madeline Gendreau[1], Ellyn Enderlin[1], and Rainey Aberle[1]

[1]Department of Geosciences, Boise State University, Boise, USA.

**Correspondence:** Jukes Liu (jukesliu@boisestate.edu)

**Abstract.** En masse application of feature-tracking algorithms to satellite image pairs has produced records of glacier surface velocities with global coverage, revolutionizing the understanding of global glacier change. However, glacier velocity records are sometimes incomplete due to gaps in the cloud-free satellite image record and failure of standard feature-tracking parameters (e.g., search range, chip size, estimated displacement, etc.) to capture rapid changes in glacier velocity. Here, we present a
pipeline for pre-processing commercial high-resolution PlanetScope surface reflectance images (available daily) and for generating georeferenced glacier velocity maps using NASA's autonomous Repeat Image Feature Tracking (autoRIFT) algorithm with customized parameters. We compare our velocity time series to the NASA ITS_LIVE global glacier velocity dataset, which is produced using autoRIFT, with regional-scale feature-tracking parameters. Using five surge-type glaciers as test sites, we demonstrate that the use of customized feature-tracking parameters for each glacier improves upon the velocity record provided by ITS_LIVE during periods of rapid glacier acceleration (i.e., change of > several meters per day over 2–3 months). We
show that ITS_LIVE can fail to capture velocities during glacier surges, but that both the use of custom autoRIFT parameters and the inclusion of PlanetScope imagery can capture the progression of dramatic changes in flow speed with uncertainties of only $\sim 0.5$ m/d. Additionally, the PlanetScope image record approximately doubles the amount of cloud-free imagery available for each glacier and the number of velocity maps produced outside of the months affected by darkness (i.e., polar night), aug-
menting the ITS_LIVE record. We demonstrate that these pipelines provide additional insights into speedup behavior for the test glaciers and recommend that they are used for studies that aim to capture glacier velocity change at sub-monthly timescales and with greater spatial detail.

## 1 Introduction

Feature-tracking in optical images is broadly used to measure the displacement of the Earth's surface over time and requires
identification of matching visual features between images collected at different times. When applied to repeat-pass satellite images, it can be used for image coregistration (Sommervold et al., 2023) or tracking of ground or ice surface motion (Van Wyk de Vries et al., 2022). We focus on its applications for tracking glacier surface velocities, although the processing method we present may be more broadly applied.

Feature-tracking in optical satellite images has been used to measure ice velocity at scales ranging from individual sites to
entire regions (Van Wyk de Vries and Wickert, 2021; Howat, 2020; Gardner et al., 2023). The NASA Making Earth System





Data Records for Use in Research Environments (MEaSUREs) ITS_LIVE dataset contains ice velocity time series for the global inventory of glaciers and ice sheets, produced using NASA's autonomous Repeat Image Feature Tracking (autoRIFT) algorithm (Gardner et al., 2023). autoRIFT is a fast, 2D normalized cross-correlation technique that can be applied to time series of optical images from Sentinel-2 and Landsat as well as SAR images from Sentinel-1 (Lei et al., 2021). In each image pair,

the second (i.e., later) image is divided into patches called chips, which are matched within a search range to the first/reference image by identifying the peak normalized cross-correlation value (Lei et al., 2021). autoRIFT iteratively tests larger chip sizes until feature displacements are successfully identified. autoRIFT applies a Normalized Displacement Coherence filter to identify and remove low-coherence displacement results, based on displacement thresholds that are scaled to the local search range (Lei et al., 2021). The additional use of autoRIFT's sister package, Geogrid, geocodes autoRIFT's displacement outputs

and allows for the creation of georeferenced velocity maps (Lei et al., 2021).

Successful feature-tracking by ITS_LIVE is facilitated by the iterative identification of the ideal chip size as described above, incorporation of all image pairs with acquisition dates $\leq$ 545 days apart, and the use of regional-scale downstream search ranges identified using reference velocities (Gardner et al., 2023). However, for glaciers with large changes in speed on seasonal to multiyear timescales, the use of static reference velocity maps results in failed feature identification if the actual

displacements far exceed the reference velocities, as shown in (Liu et al., 2024) and the examples we present here. Gaps in glacier velocity records hinder the study of a variety of speedup behavior that provide insight into internally-driven (Liu et al., 2024; Beaud et al., 2022; Benn et al., 2023; Paul et al., 2022) and externally-forced (King et al., 2020) changes in glacier velocity.

One way to increase the amount of velocity data produced through autoRIFT is to customize the search ranges and chip

sizes for each glacier, as done in Liu et al. (2024). Another strategy is to apply autoRIFT to other image products such as the ∼daily high-resolution (< 5 m) optical imagery from the PlanetScope (PS) satellite constellations. Three active PS satellite generations totalling more than 170 satellites collect visible and near infrared observations at various times throughout the day, enabling daily repeats for ∼90% of the Earth's land (Labs, 2022; Roy et al., 2021).

Although the PS images have superior temporal resolution compared to the government-owned images traditionally used

for feature-tracking (i.e., Landsat and Sentinel-2), use of these images can be challenging. Unlike the standard Landsat and Sentinel-2 products, there are inconsistencies in PS image radiometry (Leach et al., 2019; Frazier and Hemingway, 2021; Latte and Lejeune, 2020; Kington and Collison, 2022) between satellites as well as differences in local overpass times and image spatial resolution (3.0–4.2 m) across generations of satellites (Roy et al., 2021; Labs, 2022). Additionally, the relatively small footprints provided often require mosaicking of multiple images to cover the full study area, which can result in inaccurate

displacements across image mosaics due to coregistration errors (Leach et al., 2019; Frazier and Hemingway, 2021; Latte and Lejeune, 2020). Furthermore, the proportion of images with standard quality (i.e., sun altitude $\geq$ 10°, off nadir view angle less than 20°, $\leq$ 20% saturated pixels) is substantially lower in many glacierized regions because they tend to be located at relatively high latitudes where cloud and snow cover adversely impact geolocation and spectral response accuracies (Roy et al., 2021). Even for the same satellite, there may be substantial variation in the surface reflectance values between images especially in

challenging lighting conditions (i.e., low sun angles, large changes in topographic shading from steep terrain, and saturation





over bright features such as snow and glaciers) (Kington and Collison, 2022). Here, we present a pre-processing method to address these challenges and standardize the PS imagery for use in optical feature-tracking with autoRIFT using customized parameters, which we refer to as custom autoRIFT.

We apply the custom autoRIFT pipeline to Sentinel-2, Landsat 8/9, and the pre-processed PS images for five glaciers in
a wide range of geographic and climate settings that are known to have periodic, order-of-magnitude or greater variations in speed (i.e., surges) in order to assess the performance of the modified velocity-mapping pipeline. We demonstrate that the use of PS imagery, processed with customized autoRIFT parameters, may be harnessed to increase the temporal coverage of glacier velocity data during periods of rapid flow speed variability.

## 2  Methods

Here, we describe the required inputs and steps of the PS custom autoRIFT pipeline for mapping glacier velocities. We apply the pipeline to five surge-type glaciers from a range of geographic and climatic settings: Aavatsmarkbreen in Svalbard; Sít' Kusá (Turner Glacier) in Alaska; Nàlùdäy (Lowell Glacier) in the Yukon; Medvezhiy Glacier in the Pamir Mountains; and South Rimo Glacier in the Karakoram. For each glacier, we map surface velocities over a ∼2-year period and compare the results to custom autoRIFT velocities produced from Sentinel-2 and Landsat image pairs over the same time period, as well as
the ITS_LIVE velocity record.

The PS pre-processing and custom autoRIFT pipeline is summarized in Figure 1. The PS image pre-processing steps are contained in a GitHub repository named `planet_tile2img` while the custom autoRIFT steps are contained in a GitHub repository named `SK-surge-mapping`. Landsat (LS) and Sentinel-2 (S2) images cropped to the AOI can be directly input to the custom autoRIFT steps. Scripts to download and crop these other satellite images are also provided in the
`SK-surge-mapping` repository. These repositories are made available through the "Code availability" section at the end of the manuscript.

### 2.1  Inputs

The inputs for the PS custom autoRIFT pipeline include the pre-processed PS images, several geospatial files specific to the study site, and parameters for the custom autoRIFT algorithm as described below.

The three PS satellite generations achieve sub-daily temporal resolution and cover the Earth's surface from ±81.5 degrees of latitude depending on the season, with more overlap in the high latitudes due to the satellites' sun-synchronous orbits (Labs, 2022; Roy et al., 2021). PS images acquired in darkness are not processed and excluded from the databases because they cannot be geolocated, resulting in seasonal temporal gaps at high latitudes (Roy et al., 2021). For a mid-latitude study site of 20x30 km in size, for example, 200–800 GB of downloadable imagery may available for a two-year period. We use the Sentinel-2
(S2) harmonized and normalized data, which are generated by fitting a non-linear model that transforms the input PS scene's reflectance coefficients to match the S2 spectral response from an S2-based seasonal reference dataset, minimizing large deviations between scenes from differences in day and ambient conditions (Kington and Collison, 2022). The harmonization and





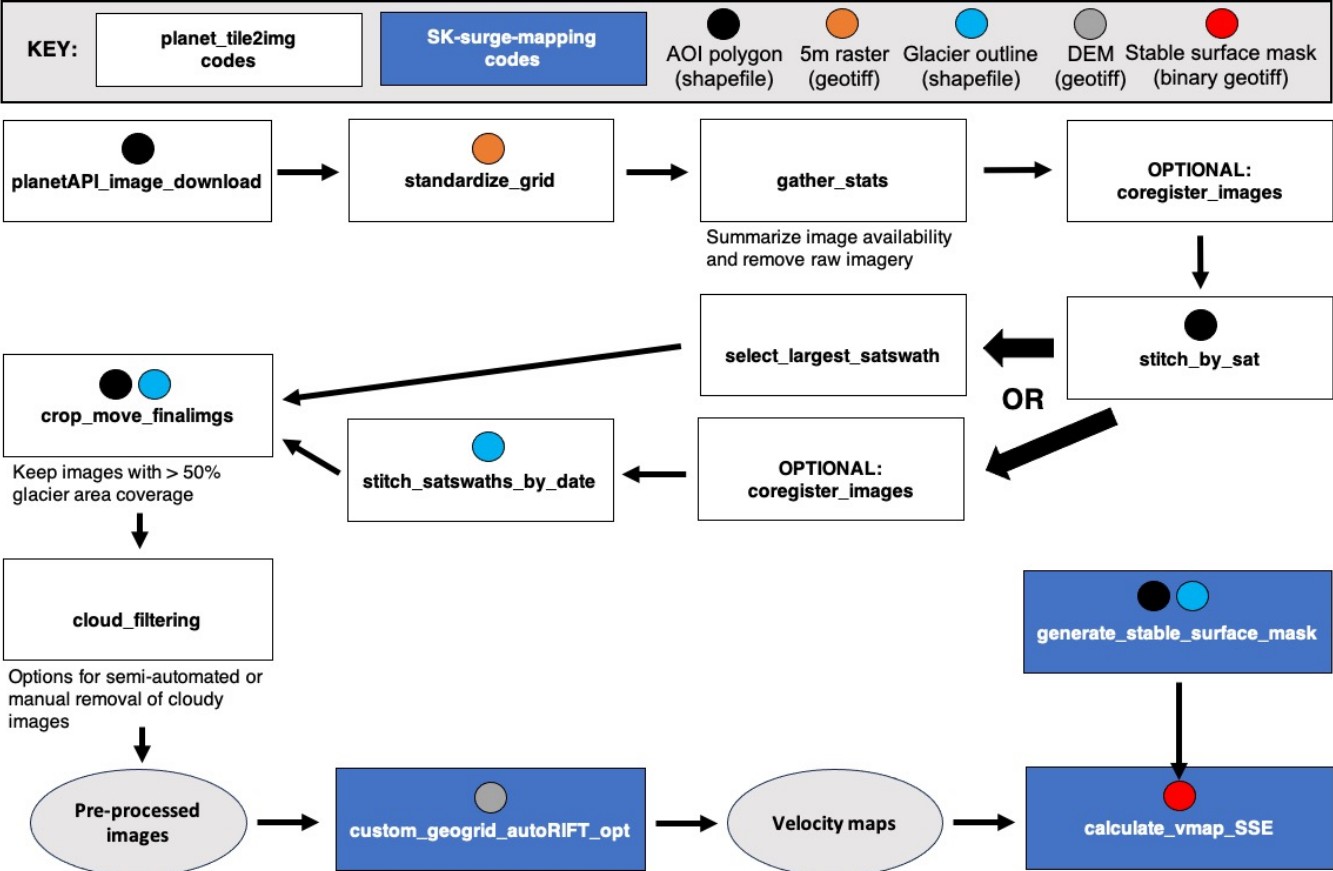

**Figure 1.** Flowchart of pipeline for processing PlanetScope imagery for input to custom autoRIFT. Each of the rectangles correspond to a Python script in either the `planet_tile2img` or `SK-surge-mapping` code repository, distinguished by rectangle color. The circles within the rectangles correspond to an input for that script. These repositories are provided in the "Code Availability" section at the end of the manuscript.

normalization reduces the differences in surface reflectance between scenes due to variation in lighting conditions, especially at low sun angles, large changes in topographic shading from steep terrain, and saturation over bright features such as snow and glaciers (Kington and Collison, 2022). The normalization of PS images by Planet Labs may deteriorate the absolute radiometric accuracy but provides increased consistency when comparing different images (Kington and Collison, 2022).

The auxiliary geospatial file inputs for each glacier site include: 1) a glacier outline shapefile, 2) an Area of Interest (AOI) shapefile, 3) a 5 m resolution geotiff covering the AOI to standardize the spatial resolution of all images, and 4) a digital elevation model (DEM) covering the AOI (Fig. 1). The glacier outline can be downloaded from the Global Land Ice Measurements from Space database with the Randolph Glacier Inventory (RGI) (Consortium, 2017) or manually delineated. The AOI shapefile should be a rectangular polygon that covers the glacier and surrounding land. All downloaded images will be



cropped to this AOI. The glacier outline and AOI can be cropped to cover only the portion of the glacier that is of interest to reduce computation time for larger glaciers. The 5-m resolution geotiff is used solely to standardize the spatial resolution and grid of all downloaded PS images. Here we use 30 m resolution Landsat surface reflectance images resampled to 5 m using the

GRASS r.resamp.rst function in QGIS, which reinterpolates the image using a regularized spline with tension and smoothing. The DEM is used to guide the downstream search in the autoRIFT algorithm and for georeferencing the output velocity map (Gardner et al., 2023). The DEMs used in this study are listed in Table 1.

Optional inputs for the custom autoRIFT pipeline include a reference velocity map and a stable surface mask. Search ranges can be adjusted manually using a scaling factor or automatically using the reference velocity map, which must be input as

separate x-direction and y-direction velocity files. A stable surface mask can be generated automatically through the pipeline using the glacier outline and AOI polygon, but may require manual editing to exclude any other bodies of ice within the AOI and outside of the glacier outline. The mask is a binary raster, where "1" values represent stable surfaces, and "0" values represent all other pixels within the AOI. The script `generate_stable_surface_mask` will automatically generate the mask as the inverse of glacier outline provided as input with the same extent as the AOI. Other glaciers and/or ice mélange

present in the AOI outside of the glacier outline must be manually removed from the stable surface mask. We manually edited our stable surface masks to remove other glacier ice within the AOIs using QGIS. The stable surface mask will mask out any velocities over static surfaces surrounding the glacier if included as an input to autoRIFT. However, we used the masks in post-processing to calculate the stable surface error (SSE) associated with each velocity map, using `calculate_vmap_SSE`, as done in Nolan et al. (2021) and Liu et al. (2024). The SSE is equal to the Root Mean Squared Error (RMSE) in speed from all

stable surface pixels according to the stable surface mask.

## 2.2 Pipeline

The PS images must first be downloaded either through the Planet Explorer or a Planet API. Image download requires a Planet Labs account and its associated API key. Free access to a Planet Labs account is provided through the NASA Commercial Smallsat Data Acquisition Program program. Here, we programmatically download S2-harmonized images (see Section 2.1)

through the Planet API. Cloudy images can be filtered before downloading using a cloud cover threshold, calculated with the provided cloud masks. We used the Near Infrared (NIR) images because the band range and width are the most consistent across the three PS satellite generations and correspond most closely with the S2 band range and width (Labs, 2022). The NIR images are downloaded as individual "tiles", which must be stitched along-track for each satellite overpass. The downloads and subsequent pre-processing steps are applied to monthly batches of images, which are split into directories named by their

date, in `YYYY-MM` format. The download step may fail if a large number of tiles are available for the month. We provide an optional half-month alternative for the download script but users must manually consolidate the directories for each month to proceed with the following steps using the existing code.

Following bulk image download, all the image tiles are resampled to the 5 m grid of the input geotiff using a linear spline interpolation in order to standardize the spatial grid (`standardize_grid`). Then, we record the number of tiles available and





satellite overpasses (i.e., swaths) each day of the month to characterize image availability for the month (`gather_stats`). Since only the resampled files are used in subsequent steps, we remove the raw files to reduce storage needs.

The resampled image tiles are stitched along-track for each satellite, producing a "satellite swath" associated with the acquisition date and PS satellite ID. The PS images are coregistered with the accuracy of 10 m RMSE at the 90th percentile (Labs, 2022) and some of the image tiles may not stitch together smoothly due to errors in image georeferencing. Thus, there is an

optional coregistration step (`coregister_images`), which performs a 2D cross-correlation of the images and finds the x and y offset that corresponds to the maximum normalized correlation value. For each pair of overlapping image tiles, if the correlation value is above the correlation threshold (default 0.8) and the x and y offset are less than the offset threshold (default 2 pixels, or 10 m), then the second image will be shifted accordingly (Fig. 2). All coregistration thresholds can be adjusted manually within the pipeline.

In the case where multiple satellite swaths overlap on a single day, either one swath can be selected or the overlapping swaths can be mosaicked to increase spatial coverage. Here we mosaicked all satellite swaths for each day in order to maximize the spatial coverage of the imagery using `stitch_satswaths_by_date`. The acquisition time for overlapping swaths can differ by seconds to hours (Roy et al., 2021), which can impact the consistency in brightness between the images, even after pre-processing. Within the region of swath overlap, the mean brightness difference is used to uniformly adjust the brightness

values in the smaller swath in order to minimize brightness bias between swaths. Another option available through the pipeline is to select one swath per day using `select_largest_satswath`. By default, the swath with the greatest spatial coverage within the AOI is selected if only using a single swath for a date. Alternatively, the user can manually select the best swath by copying its file into the directory containing the final images for each date. Figure 2 demonstrates the process for both scenarios: selecting a single satellite swath versus stitching multiple satellite swaths.

Once the single satellite swath or mosaicked swaths for each date are selected, the final images are cropped to the AOI and the images with greater than 50% data coverage over the glacier area are automatically moved into a separate directory, through the `crop_move_finalimgs` script. To optimize computation and analysis time, images with clouds covering part or all of the glacier may then be removed. Clouds can be filtered manually or automatically using the script provided (`cloud_filtering`), which plots the images to aid in manual filtering but also includes optional steps to test automated

filtering methods that rely on gradients in pixel brightness. For the example applications presented herein, we manually filtered cloudy images because the cloud brightness varies widely between sites. Even for images captured at the same site, cloud brightness thresholds need to be calibrated carefully.

In addition to the final, non-cloudy images, the required inputs to custom autoRIFT include a DEM over the AOI. The DEM used for each test site is listed in Table 1. Within the `custom_geogrid_autoRIFT_opt` script, the user must

input the chip sizes, which controls the spatial resolution of the resulting velocity maps, and minimum and maximum date separations. The input of a reference velocity map automatically sets search ranges accordingly, which is particularly helpful during speedups and for glaciers where there are large spatial differences in speed. In the absence of a reference velocity map, one may manually test different search ranges within the script. The modification of the search range within the script imposes a static search range value to the entire AOI. In this study, a reference velocity map during a glacier surge was only available





**Figure 2.** Process for constructing final images over Sít' Kusá for two acquisition dates, one with multiple satellite overpasses and one with a single overpass. Axes labels are in pixels. The satellite swaths are labelled by acquisition date and 4-digit satellite ID. The number of tiles corresponding to each swath are listed in the corner of each swath image. The tiles for the 2020-03-12 swaths are not shown due to space limitations. The red triangle in the 2D cross correlogram associated with the optional coregistration step indicates the offset corresponding to the maximum normalized correlation.

for Sít' Kusá. We used the velocity map from during the 2006 surge from Nolan et al. (2021) as the input. For the other sites, we applied large search ranges of 16x the input chip size.

### 2.3 Other velocity data

Sentinel-2 and Landsat 8/9 images may also be downloaded and input to the custom autoRIFT pipeline. The `SK-surge-mapping` repository provides scripts to automatically download Sentinel-2 Cloud Optimized Geotiff (COG) images and Landsat 8/9 Col-

lection 2 Level 1 images through Amazon Web Services (AWS), through Planet Explorer, or by adjusting the `planetAPI_image_download` script. We downloaded the 10 m resolution NIR band from Sentinel-2 images to match that of the PS imagery. For Landsat 8/9,



the 15 m-resolution panchromatic band was used because it has the highest spatial resolution of all Landsat bands. All images were cropped to the AOI and then filtered for cloud coverage using the same `cloud_filtering` step as the PS images. These cropped images can be directly input into custom autoRIFT.

The ITS_LIVE velocity time series can be extracted for specific point locations using the ITS_LIVE widget (Gardner et al., 2023). The time series include velocities from standard autoRIFT parameters (chip sizes and search ranges determined for an entire region) applied to Sentinel-1, Sentinel-2, and Landsat imagery. The error associated with each ITS_LIVE velocity measurement is calculated from the standard deviation of velocities measured over a stable surface after applying a geolocation offset (Gardner et al., 2023). If the velocity field does not intersect a stable surface, the error is assigned as the residual sum of 185   squares of the satellite pointing uncertainty (i.e., positioning and orientation accuracy) for both images (Gardner et al., 2023).

## 2.4   Test Applications

We chose five glaciers to assess the performance of the custom autoRIFT pipeline relative to the ITS_LIVE dataset (Fig. 3). We focused specifically on surge-type glaciers because they represent particularly challenging conditions for feature-tracking given their one or more order-of-magnitude variations in velocity. We included five surge-type glaciers from varying climate 190   regimes and spanning a wide range of latitudes (35–79 °N) to account for differences in image quality and quantity in pipeline performance. Three of the study glaciers have surged since 2019: Sít' Kusá (Turner Glacier), Nàlùdäy (Lowell Glacier), and South Rimo Glacier. Sít' Kusá was confirmed to have surged from 2020–2021 using in situ and remotely-sensed velocities (Liu et al., 2024). South Rimo Glacier surged in 2019 (Jiang et al., 2021) and Nàlùdäy surged in winter of 2023 (Van Wychen et al., 2023) based on inspection of remotely-sensed velocities. Table 1 summarizes the previously-measured quiescent and surge 195   flow speeds that are used to guide the selection of minimum and maximum date separations used for velocity map processing. The glaciers with continental climates include Nàlùdäy/Lowell Glacier (LO) on the Canadian side of the St. Elias mountain range, Medvezhiy Glacier (MZ) in the Pamir mountains, and South Rimo Glacier (SR) in the Karakoram. The glaciers with maritime climates include Sít' Kusá (SK) on the Alaskan side of the St. Elias mountain range and Aavatsmarkbreen (AV) on Spitsbergen, Svalbard.

**Table 1.** Digital Elevation Models (DEMs) used for study site analysis and other glacier site characteristics, e.g. quiescent and surge flow speeds. Flow speeds are from the studies listed under "speed source".

| Site | Latitude | DEM source | DEM resolution | Quiescent speed | Surge speed | Speed source |
|------|----------|------------|----------------|-----------------|-------------|--------------|
| AV | 78.64°N | GMTED2010 | 51.51 m | ∼0.5-0.6 m/d | 4.5 m/d | Sevestre et al. (2018) |
| SK | 60.06°N | IfSAR Alaska | 4.97 m | <1-8 m/d | 25-30 m/d | Liu et al. (2024); Terleth et al. (accepted) |
| LO | 59.97°N | ArcticDEM | 9.65 m | <1 m/d | 30 m/d | Bevington and Copland (2014) |
| MZ | 38.65°N | SRTM | 26.00 m | ∼1.5 m/d | 100 m/d | Dolgushin et al. (1974); Raymond (1987) |
| SR | 35.32°N | GTOPO30 | 26.85 m | 0.3 m/d | 12 m/d | Jiang et al. (2021) |

We mapped velocities for each test glacier over a 2-year period in order to capture seasonal velocity changes for at least one full season and the full surge for each of the glaciers with confirmed surges. The temporal resolution of the velocity estimates



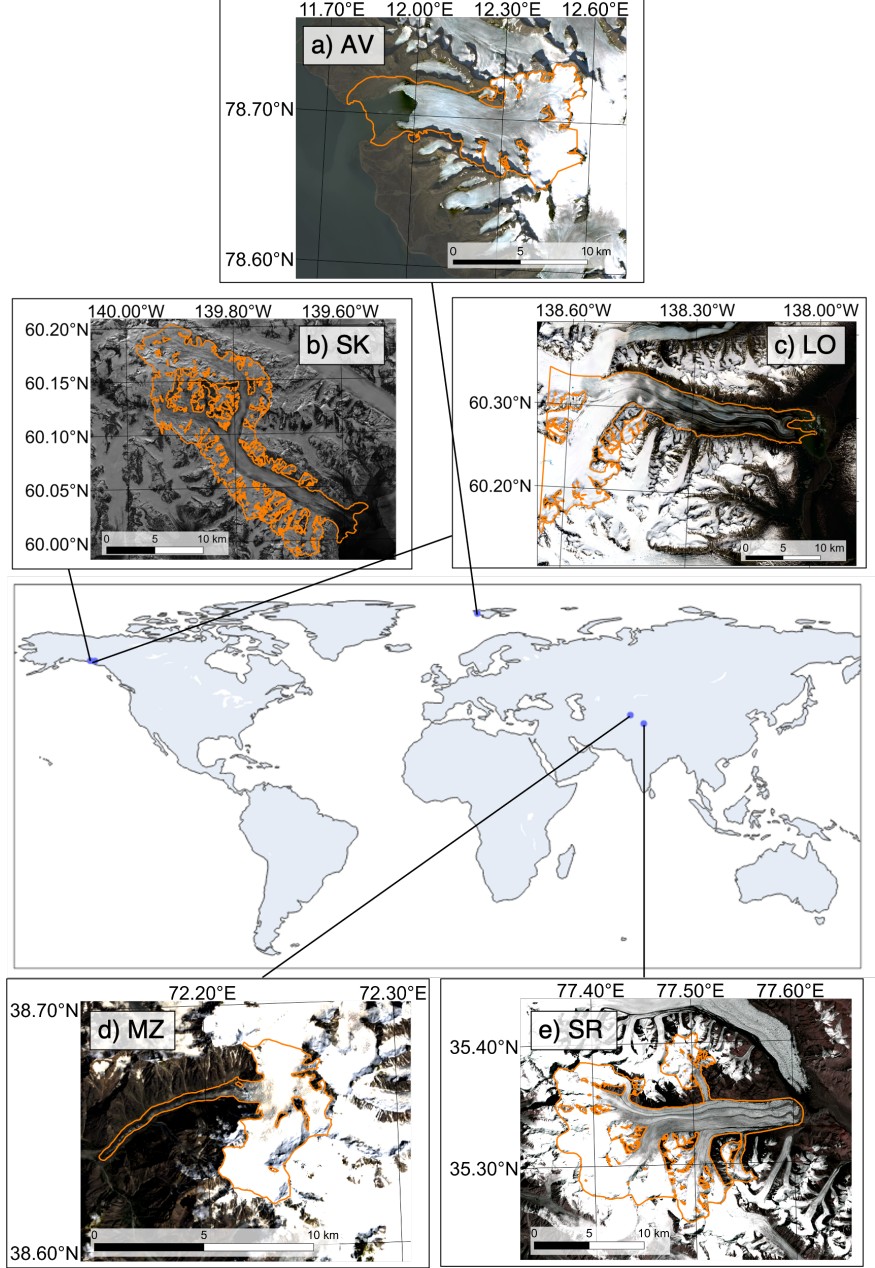

**Figure 3.** The study sites are five surge-type glaciers from a range of climatic and geographic settings: a) Aavatsmarkbreen in Svalbard, b) Sít' Kusá in Alaska, c) Nàlùdäy in the Yukon, d) Medvezhiy Glacier in the Pamirs, and e) South Rimo Glacier in the Karakoram. Background images for a)–d) are from Landsat 8/9. The RGI glacier outlines (orange) may not represent each glacier's current extent.

varied between glaciers based on their flow speeds and image availability. Velocities for glaciers with fast flow ($\geq$ 5 m/d)—SK,





LO, and SR—were mapped using image pairs with date separations of 5 to 60 days, as done successfully for SK's surge in Liu
et al. (2024). Velocities for glaciers with slow flow speeds ($\leq$ 5 m/d)—MZ and AV—were mapped with longer date separations
of 14 to 60 days in order to successfully detect displacement across the minimum chip sizes and search ranges for autoRIFT.
For the PS images, we automatically mosaicked all the satellite swaths for each day. For the sake of computational efficiency,
the optional across-swath coregistration was only applied to Medvezhiy Glacier because its small size and relatively slow flow
speeds make it susceptible to velocity errors from poor geolocation (Table 1). We compared the temporal coverage and quality
of the PS, Sentinel-2, and Landsat 8/9 custom autoRIFT velocity maps produced and assess how they contribute to the velocity
record provided through ITS_LIVE.

## 3   Results

Here, we summarize the usable image coverage, the quality and characteristics of the custom autoRIFT velocity maps, and
their improvements to the velocity record provided through ITS_LIVE for the test sites over a ~2 year period. Since glacier
flow can vary considerably over seasonal and shorter time periods, we take into account variations in image coverage across
boreal winter (Dec., Jan., Feb.), spring (Mar., Apr., May), summer (Jun., Jul., Aug.), and fall (Sep., Oct., Nov.) seasons.

In general, higher latitude corresponded to lower image coverage centered around the boreal winter solstice (Dec. 21st/22nd).
Figure 4 shows the availability of non-cloudy images available for PlanetScope (PS), Sentinel-2 (S2), and Landsat 8/9 (LS) for
a two-year period. SR, located in a continental climate and at the lowest latitude of $\sim 35°$N, had the most seasonally consistent
record of imagery – at least 18 PS or S2 images for each three-month season and the greatest number of winter images: 67
for PS, 41 for S2, and 9 for LS (Fig. 4a). MZ is situated only 3 degrees of latitude higher and had a similar image availability
to that of SR. MZ had a denser PS and LS image record (247 and 50 images, respectively) overall, but slightly fewer winter
images (63 PS images) (Fig. A1). Due to its similarities with SR, its barcode plot is located in the supplement. Out of all the
sites, velocity maps for the two lowest latitude sites, MZ and SR, provided the most consistent velocity coverage more than
$\pm 1$ month from the winter solstice (Fig. 5). For the sub-Arctic sites, LO and SK, there were fewer images (< 15) each winter
because of the longer period of total darkness (i.e., polar night) and/or deep shadows due to low sun angles. Meanwhile, the
highest-latitude site, AV, is located in the high Arctic where polar night obscures optical image collection for nearly half the
year (from mid-September to March) (Fig. 4b). However, during periods of solar illumination (March to mid-September), the
increase in satellite track overlap at higher latitudes resulted in a denser record of images available for velocity mapping in
comparison with the sub-Arctic sites (Figs. 4,5a-c).
The sites in maritime climates had lower image coverage due to more frequent cloud cover than continental sites. Although
the two sub-Arctic sites, LO and SK, are situated at the same latitude, LO is located in a continental climate zone whereas SK
is located in a maritime climate and has more frequent cloud cover, which reduces the number of usable images by one-third to
one-half (Fig. 4). However, the image records for PS and S2 often complemented each other: some S2 images were available
in PS image gaps (e.g., 2020 spring and 2021 spring) and some PS images were available in S2 image gaps (e.g., 2020 fall and
2021 fall) (Fig. 4d).



**Figure 4.** Barcode plots showing cloud-free image availability for the different satellite platforms for four of the study sites over two years. See supplemental figure A1 for the barcode plot for MZ. Left column (a,c) contains sites in continental climates where the right column (b,d) contains sites in maritime climates. The top panel for each site corresponds to the number of tiles and satellite overpasses from the PlanetScope satellites. Red and gray boxes along the bottom and top of the barcodes mark the winter and summer months, respectively. Counts for each season are listed along the barcode and the total count is shown to the right of the barcode.



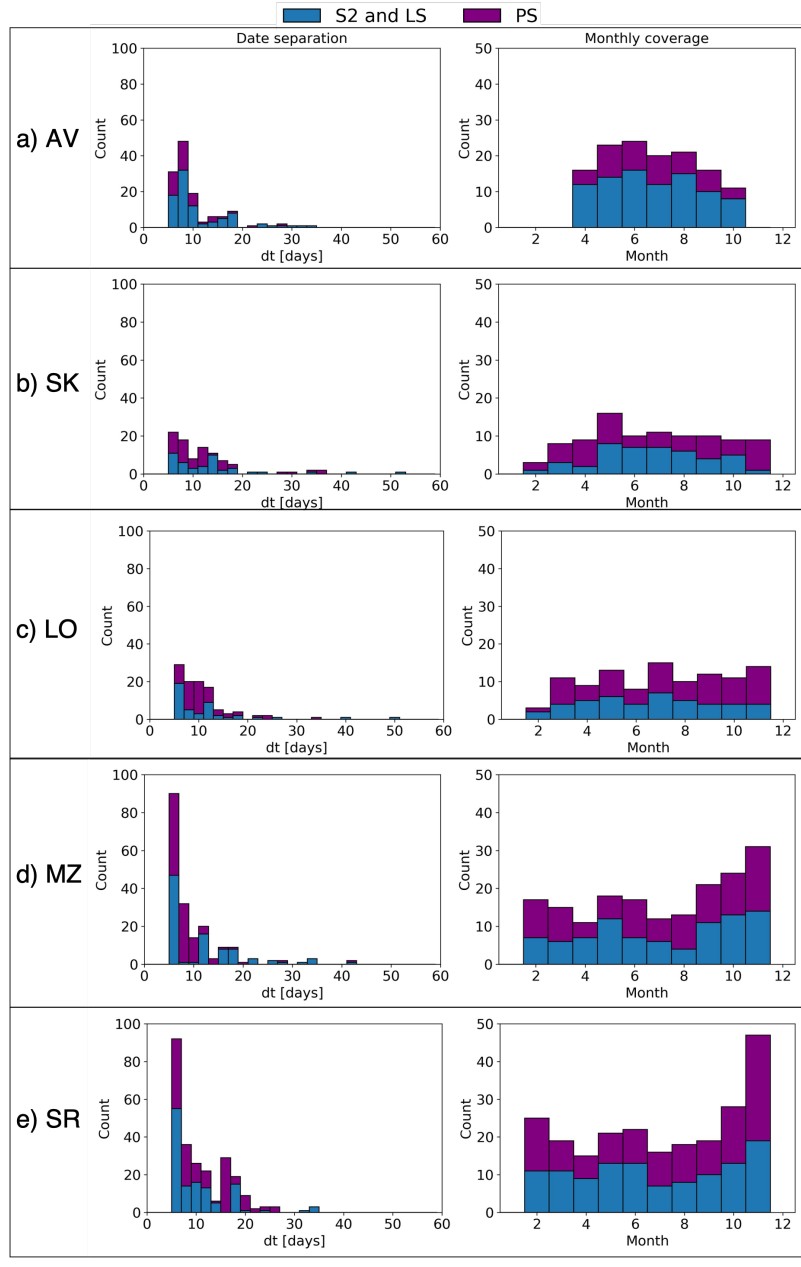

**Figure 5.** Histograms comparing date separations and monthly coverage for velocity maps from PlanetScope (PS) versus Sentinel-2 (S2) and Landsat (LS) image pairs with date separations between 5 and 60 days for each of the study glaciers. Sites are displayed in order of latitude from north to south.

The addition of PS imagery approximately doubled the amount of velocity maps produced (Fig. 5) outside of the Arctic, where the S2 and LS coverage was already higher in the summer months unaffected by polar night (Figs. 4b & 5a) due to





the increase in orbit overlap at high northern latitudes (Li and Roy, 2017). For the sites other than AV, the temporal resolution provided by the PS imagery was 1.1–3.0x greater than the coverage provided by S2 imagery, and for SR, MZ, and SK, 1.1–1.8x

greater than the coverage provided by both S2 and LS imagery combined (Fig. 4a,c,d; Fig. A1). Notably, the addition of PS imagery increased the number of velocity maps with date separations shorter than 20 days (Fig. 5), which may be advantageous during times of fast flow. For MZ and SR, there were date separations of < 20 days that were only achieved with PS image pairs (Fig. 5d,e). In general, the addition of PS velocity maps produced more uniform monthly velocity coverage, especially in November (Fig. 5b–e).

The custom autoRIFT pipeline successfully increased the temporal coverage of velocity observations compared to the ITS_LIVE record and captured higher maximum flow speeds for the four test sites with pronounced seasonal and/or interannual variability. Gaps in the ITS_LIVE record were common during periods of ice acceleration (e.g., Fig. 6b,d,f and Fig. 7d), which often occur when glacier speeds increased above 5 m/d (Fig. 6b,d,f). There were gaps during periods of acceleration even when speeds were less than 5 m/d (Fig. 7d). Figures 6 and 7 show the ITS_LIVE velocities produced from optical satellite

image pairs only, but the equivalent plots that include the Sentinel-1 SAR velocities (shown in figure A2) corroborated these observations.

For SK, ITS_LIVE failed to capture velocities from May 2020 through March 2021 (Fig. 6b). In contrast, the custom autoRIFT velocities captured the full progression of the 2020–2021 surge for the near-terminus area, tracking the acceleration from <1 m/d to 20+ m/d (Fig. 6b). The additional velocity data revealed the seasonality of velocity change throughout the

surge linked to the evolution of glacier hydrology (Liu et al., 2024). Notably, the inclusion of PS velocity maps captured critical velocity data through October/November 2020 and January/February 2021, filling gaps in the other velocity records during periods of darkness (Fig. 6b). The additional PS velocities confirm that winter velocities remained elevated throughout the SK's surge compared to quiescent velocities, as detailed in (Liu et al., 2024). At SR, the ITS_LIVE velocities only captured a peak velocity of ∼5 m/d near the glacier terminus (Fig. 6c,d). LS and PS velocities fill the ITS_LIVE gap from August to

October 2019, revealing that velocities reached a peak of nearly 10 m/d in August 2019 (Fig. 6d), closer to the results derived from SAR feature-tracking in (Jiang et al., 2021). For LO, ITS_LIVE velocities are sparse from January through August 2022 over the southern branch of the terminus, where the speedup appears to initiate (Fig. 6e,f). The custom S2 and LS velocities capture the rise to the velocity peak and the subsequent decrease, indicating the velocities reached between 15–20 m/d (Fig. 6f). Similarly at MZ, the custom autoRIFT record captures additional portions of the rise and fall from the velocity peak in the

main trunk, particularly from April to September 2021 (Fig. 7c,d). At AV, the ITS_LIVE velocity record is quite dense due to the orbit overlap at high latitudes. Because glacier velocities appear to remain steady throughout the study period at < 1 m/d (Fig. 7a,b), the standard autoRIFT parameters proved successful and the custom autoRIFT pipeline does not greatly improve the velocity record for this glacier.

For all study sites, custom autoRIFT achieved median stable surface errors (SSEs) between 0.17–0.68 m/d for all image

products, compared to the maximum errors of 1.8–3.9 m/d from ITS_LIVE. The median SSE for the custom autoRIFT velocity maps from all the sites was under 0.5 m/d, besides the median LS SSE of 0.68 m/d for AV (Fig. 8). The median SSEs from PS velocity maps ranged from 0.22–0.39 m/d, which is similar to the median SSEs from S2 (0.17–0.45 m/d) and lower than the





**Figure 6.** (Left column) Example velocity maps produced with the custom autoRIFT pipeline applied to PlanetScope (PS), Landsat (LS), and Sentinel-2 (S2) image pairs and (right column) surface speed time series at a point on each glacier marked by the white triangle at SK (a,b), LO (c,d), and SR (e,f). Error bars correspond to the stable surface error for the custom autoRIFT data and the provided error values for the ITS_LIVE data. Date separations of 5 to 60 days were used.

median SSEs from LS for all sites besides SR (Fig. 8). The SSEs from our custom velocity maps were comparable to the SSEs for ITS_LIVE (Fig. 8) despite the use of different stable surface masks between datasets.



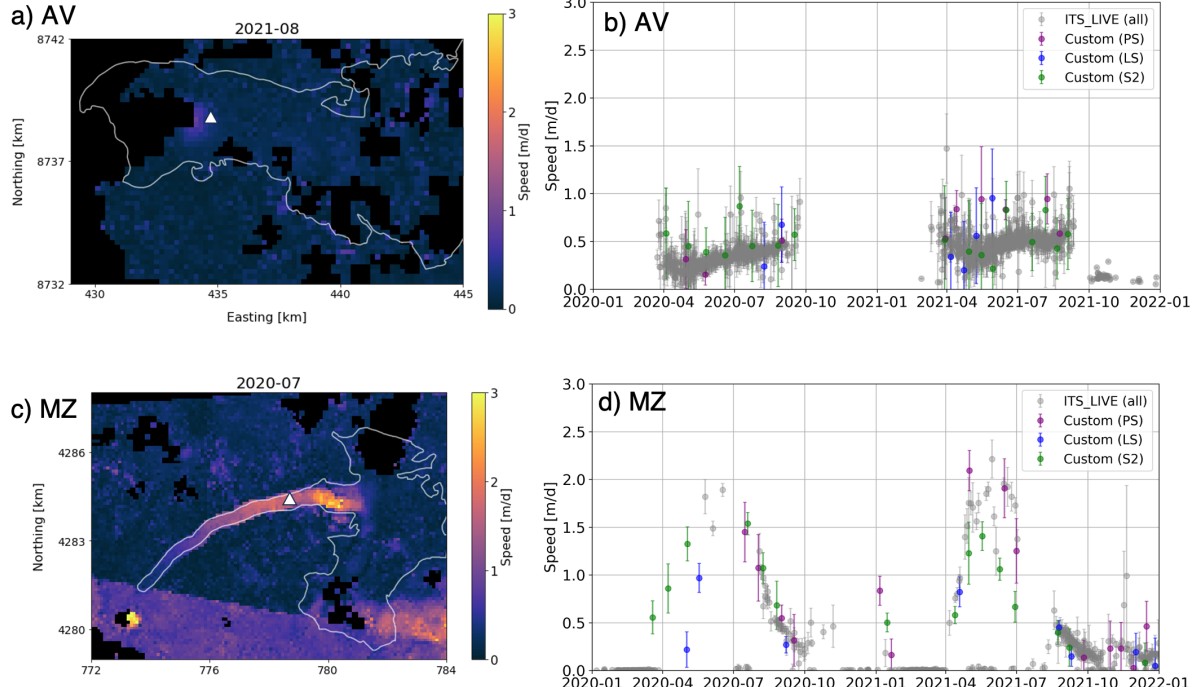

**Figure 7.** (Left column) Selected velocity maps produced with the custom autoRIFT pipeline applied to PlanetScope (PS), Landsat (LS), and Sentinel-2 (S2) image pairs and (right column) surface speed time series at a point on each glacier marked by the white triangle at AV (a,b) and MZ (c,d). Error bars correspond to the stable surface error for the custom autoRIFT data and the provided error values for the ITS_LIVE data. Date separations of 14 to 60 days were used.

## 4   Discussion

Our results show that custom autoRIFT applied to the same datasets used to create the ITS_LIVE velocity record (S2 and LS images) is needed to capture large variations in glacier velocity. The addition of PS images to custom autoRIFT enhances the temporal coverage and resolution of glacier velocity records, yielding accurate velocity maps with high spatial resolution.

For glacier sites that are seasonally velocity data-limited due to the effects of polar night, the use of PS images can extend the temporal coverage (Fig. 4d) and more than double the number of velocity maps more than $\pm 1$ month from the winter solstice, especially in the fall months (Fig. 5). The inclusion of PS imagery can also increase the velocity map pairings possible at shorter time separations (Fig. 5), which is advantageous for capturing fast glacier flow. Furthermore, custom autoRIFT can successfully measure velocities during temporal gaps in ITS_LIVE velocities that can occur near the peaks of speedups, typically above speeds of 5 m/d (e.g., in Figs. 6 &  7d). Thus, the use of both PS imagery and custom autoRIFT can be particularly useful for capturing speedups that initiate in the fall (Abe and Furuya, 2015; Nanni et al., 2023; Liu et al., 2024), as seen in the SK and SR velocity time series (Fig. 6b,d). While some winter gaps may be filled with Sentinel-1 SAR imagery, temporal



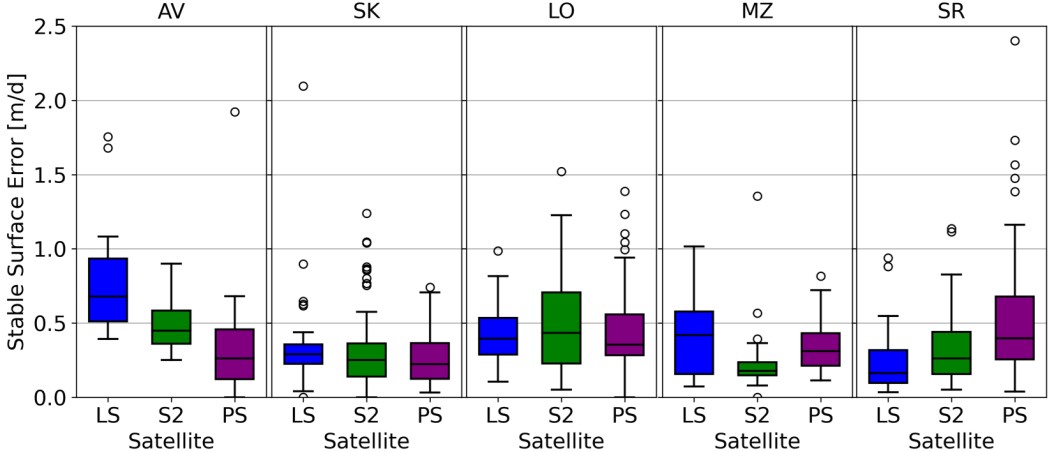

**Figure 8.** Box plots showing the calculated stable surface error of velocity maps generated with custom autoRIFT using satellite image pairs different platforms (LS = Landsat, S2 = Sentinel-2, PS = PlanetScope) for each glacier.

variations in backscatter due to the presence of liquid water prevent its use when and where melt occurs (Marin et al., 2020). Thus, the use of PS images and custom autoRIFT may be especially advantageous for glaciers in temperate climate settings where large seasonal velocity fluctuations driven by meltwater input (Burgess et al., 2013; Fountain and Walder, 1998) cause

speeds to peak at >5 m/d and where liquid water content on the surface is more common. Furthermore, for glaciers in maritime climatic settings that are frequently obscured by cloud cover, the use of PS images can improve the temporal coverage of velocity records, as demonstrated by the comparison of image availability for LO vs. SK (Fig. 4). The inclusion of PS velocity maps can also provide more uniform monthly coverage (Fig. 5b,c). In including PS imagery, however, one must be mindful of some of the artifacts in some of the PS velocity maps due to errors in coregistration and surface reflectance normalization that

remain after pre-processing, which show up as seam lines in the velocity maps (e.g., Fig. 6c and Fig. 7c). If these boundaries overlap the glacier area, the ice velocities in the overlapping area could be bias-corrected by subtracting the velocity offset in the surrounding stable surfaces. Although stitching multiple swaths can increase spatial coverage, a single satellite swath over the glacier could also be used to reduce errors associated with these boundaries. Both options can be tested within the pipeline to determine which method best balances velocity accuracy and spatial coverage.

In addition to filling temporal gaps, the use of high-resolution PS images to generate glacier velocity maps potentially resolves greater detail in velocity structure relative to S2 and LS velocity maps. Figure 9 illustrates differences in PS and S2 velocity maps from approximately the same time periods. In the case of SK, the PS velocity map showed faster velocities in the upper main trunk and tributaries of the glacier compared to the S2 velocity map, both from late July 2021 (Fig. 9e). The S2 map showed flow speeds near 0 m/d in the northern arm of the glacier whereas the PS map showed flow speeds of ∼5 m/d in

the same area (Fig. 9a,c). GPS data collected throughout field campaigns on SK during its 2020–2021 surge (Liu et al., 2024; Terleth et al., accepted) confirm that surface speeds in the main trunk were ∼5 m/d during July 2021 (Fig. A3). Thus, the PS



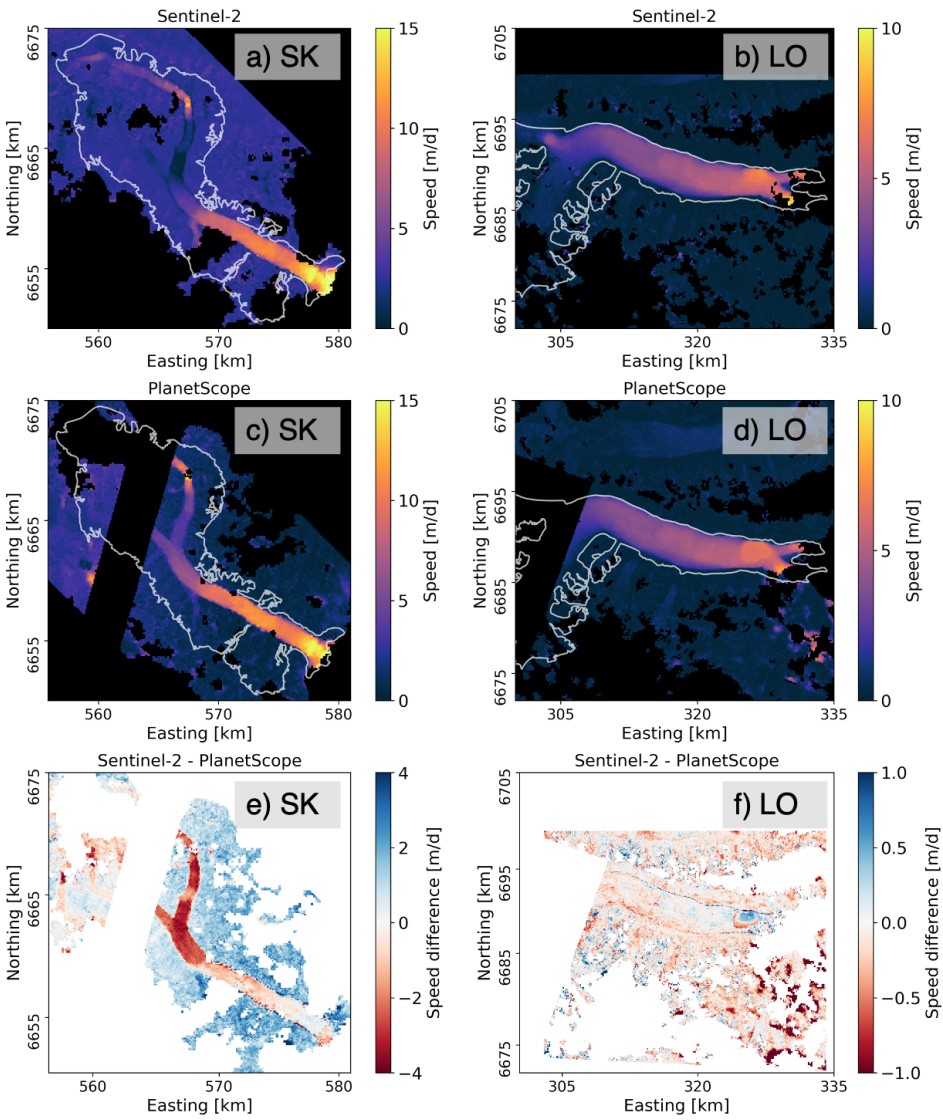

**Figure 9.** Comparison of detail in velocity maps over SK (a,c) and LO (b,d) produced from Sentinel-2 (a,b) and PlanetScope (b,d) image pairs. Image pairs have approximately the same dates for each comparison (a) 2021-07-29 to 2021-08-03 versus (b) 2021-07-30 to 2021-08-04; (c) 2022-07-26 to 2022-08-07 versus (d) 2022-07-26 to 2022-08-07. PlanetScope speeds are subtracted by Sentinel-2 speeds to calculate speed differences.

velocity map captures a more accurate representation of the velocity of structure of SK. The SSE was also higher for the S2 velocity map, supporting our interpretation that the PS velocity map provides more accurate glacier velocities in this case. In the case of LO, the speed differences between S2 and PS were <0.5 m/d throughout the glacier (Fig. 9f). The greatest velocity deviations occured where the glacier splits into a northern and southern portion of its terminus as it flows around a nunatak





(Fig. 9f). There, the PS velocity map provided more continuous spatial coverage of velocities through the two split portions, likely due to the improvements in tracking smaller features provided by the finer spatial resolution in PS images. The large amount of shear strain in this region ($\sim 3\ \mathrm{yr}^{-1}$) may rapidly deform surface crevasses and produce new fractures, as seen in the satellite images, which could hinder accurate feature-tracking in that area. In both the SK and LO examples, S2-PS differences were largest along the glacier margins (Fig. 9e,f), since feature-tracking with the higher-resolution PS imagery resolves flow closer to the glacier-land margin.

The improvements to velocity records provided by the inclusion of PS imagery and the custom autoRIFT pipeline may yield further insight to rapid changes in glacier dynamics, e.g., to the mechanisms that drive glacier speedup, as in Liu et al. (2024). Additionally, feature-tracking with the high-resolution PS images may provide more accurate and continuous estimates of velocity structure across the glacier surface. The use of custom parameters in autoRIFT increases the number of velocity maps successfully produced from different sources of optical satellite imagery. These pipelines may be particularly useful for capturing events where dynamic change occurs on monthly or shorter timescales, such as the response to drainage of a supraglacial lake, or on smaller portions of ice (e.g., tributary glaciers) that would benefit from the finer spatial resolution of PS imagery and flexible displacement search parameters.

## 5 Conclusions

We present a pipeline to pre-process daily, high-resolution PlanetScope surface reflectance images and to incorporate the images into the customized autoRIFT feature-tracking algorithm for mapping glacier velocities. The pre-processing pipeline performs additional image normalization and coregistration to allow for more seamless mosaicking of image tiles and satellite swaths. The PlanetScope images, along with optical satellite images from Landsat and Sentinel-2, may be used as inputs to the custom autoRIFT pipeline with adjustable search parameters and date separations. Using five surge-type glaciers as test sites, we demonstrate that the velocity maps produced with custom autoRIFT parameters, along with the inclusion of PlanetScope imagery, can be used to fill gaps in the ITS_LIVE velocity record which are common when flow speeds rapidly change. The inclusion of PlanetScope imagery approximately doubles the number of velocity maps and is particularly useful for filling gaps in the velocity record for glaciers that are frequently cloud-covered. The velocity maps produced with PlanetScope image pairs have stable surface errors that are less than 0.5 m/d, which are smaller than the the stable surface errors from the Landsat and Sentinel-2 custom autoRIFT velocity maps and comparable to the reported ITS_LIVE errors. Furthermore, PlanetScope velocity maps are capable of resolving more continuous velocity structure and with greater spatial detail, including large lateral variations in velocity near the glacier-land margins. The customization of autoRIFT parameters and inclusion of PlanetScope imagery can improve the velocity data coverage from the ITS_LIVE record, especially for glaciers with pronounced seasonal and/or interannual variations in velocity. Both the custom autoRIFT pipeline and PlanetScope image pre-processing pipeline produce glacier velocity data that are useful for capturing dynamic change on shorter timescales and over smaller spatial scales. We provide these pipelines as public GitHub repositories for open use and adaptation.



## Appendix A: Supplementary Material

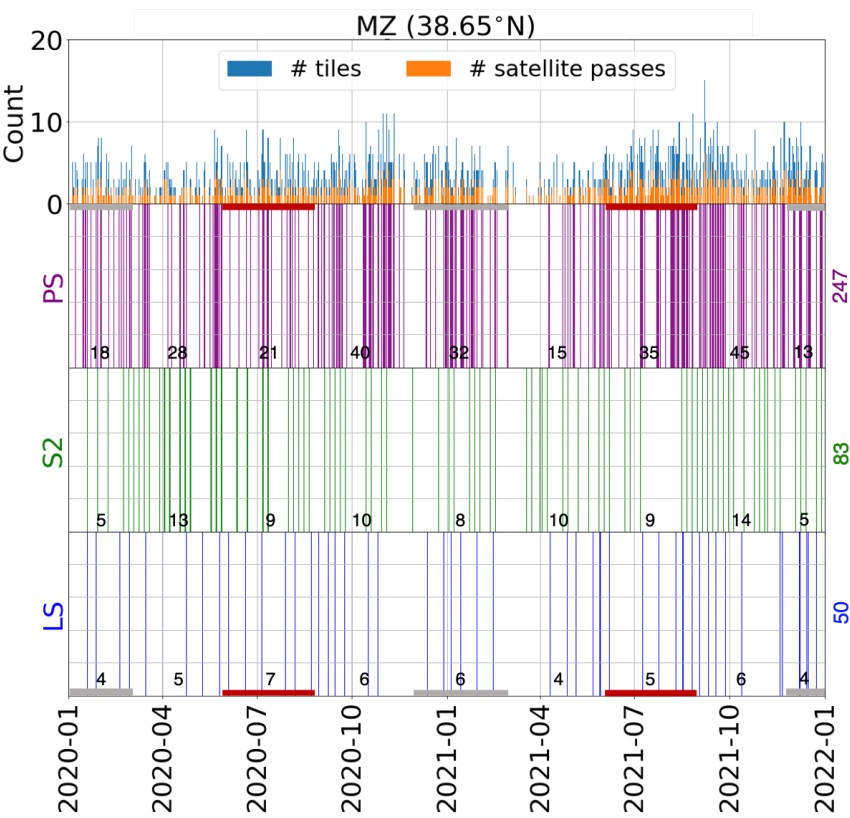

**Figure A1.** Barcode plot showing cloud-free image availability over Medvezhiy Glacier (MZ) for the three satellite platforms: PlanetScope (PS), Sentinel-2 (S2), and Landsat (LS). Red and gray boxes along the bottom and top of the barcodes mark the winter and summer months, respectively. Counts for each season are listed along the barcode and the total count for the 2-year period is shown to the right of the barcode.





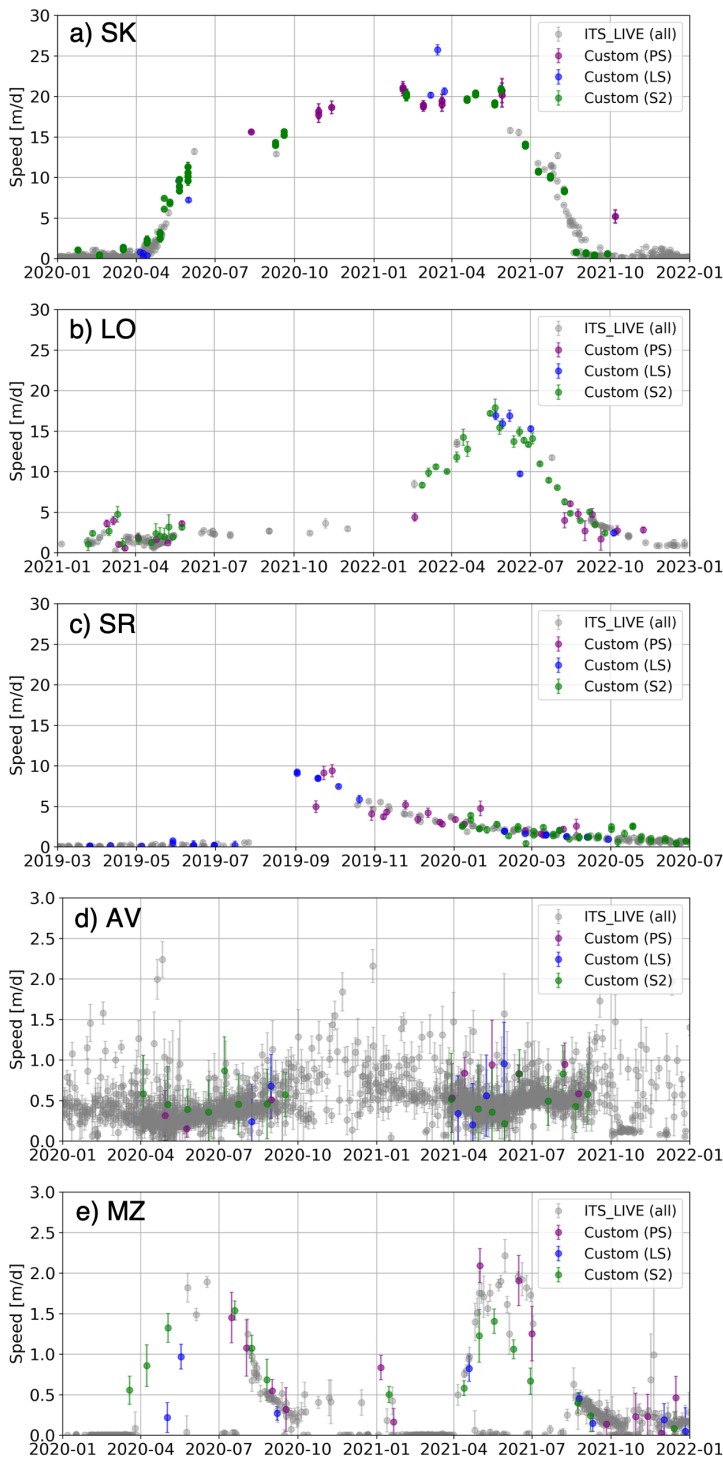

**Figure A2.** Surface speed time series including Sentinel-1 ITS_LIVE data at the points on each test glacier shown in Figures 6 and 7. a) SK, b) LO, c) SR, d) AV, and e) MZ.



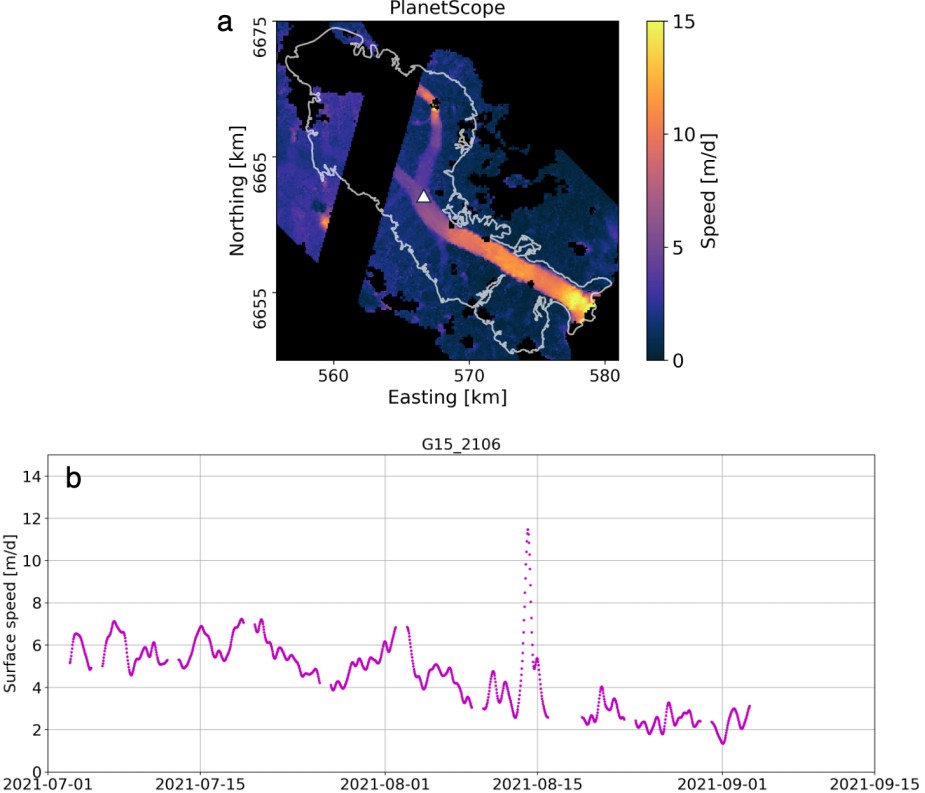

**Figure A3.** Hourly surface speeds calculated using Prescise Point Positioning processing of positions from a Trimble NetR9 GPS at site G15_2106 on Sít' Kusá from Fall 2022 indicating motion of up to ∼5 m/d in the upper main trunk of the glacier, 15 km from the glacier terminus (location marked at the white triangle in panel a).

*Code availability.* The PlanetScope image pre-processing pipeline is released on GitHub as `planet_tile2img` (https://doi.org/10.5281/
zenodo.10632745) and the custom autoRIFT code is included on the `SK-surge-mapping` GitHub repository: https://doi.org/10.5281/
zenodo.10616628.

*Author contributions.* JL, MG, and RA contributed code to the PlanetScope image pre-processing pipeline. MG and JL collected and managed the data. JL and EE designed the study and defined the scope and applications. JL wrote the manuscript and generated the figures. All authors contributed to the revision of the manuscript and figures.

*Competing interests.* The author(s) declare(s) that there is no conflict of interest regarding the publication of this article.



*Acknowledgements.* This research is funded by NASA FINESST Award 80NSSC21K1640, NSF Award ANS1954006, and the Department of Defense SMART Scholarship.



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
