# Peer review of "Improved records of glacier flow instabilities using customized NASA autoRIFT (CautoRIFT) applied to PlanetScope imagery"

_EGUsphere, 2024_

## Author Comment (AC2)

We thank both reviewers for their valuable feedback on the manuscript. The referee comments are reproduced in black text and our responses are listed in blue. One overarching change that we made was the abbreviation of "custom autoRIFT" as CautoRIFT throughout the manuscript.

**Review 1**

I have reviewed the paper entitled 'Improved records of glacier flow instabilities using customized NASA autoRIFT applied to PlanetScope imagery' submitted for consideration for publication in The Cryosphere. Using five test glaciers, this manuscript presents improvements to optically derived glacier velocities using a modified version of the NASA autoRIFT and the inclusion of PlateScope (PS) imagery. Overall, enjoyed reading this manuscript and congratulate the authors on a well written and clearly presented manuscript. The authors provide a detailed description of the processing steps, and their test cases clearly demonstrate the improvement in terms of the temporal density of velocity estimates that can be used to understand variations in glacier flow for surge-type glaciers. The only major comment that I have to improve this manuscript is to encourage the authors to include a comparison between the PS and in situ derived velocities. For example, Figure A3 presents in situ derived GPS displacements, which seem to match up with at least some of the PS derived velocities. This would provide a nice comparison/documentation of how well the PS derived velocities perform in very fast flowing situations. Other than this, I only have minor comments listed below to be addressed.

We agree that a direct comparison of the PS and in situ GPS-derived velocities available for Sít' Kusá would be valuable. We will compare the velocity maps that overlap temporally with the GPS record shown in Figure A3 and add that to the figure and discussion.

L2/3: 'glacier velocity records are sometimes incomplete due to gaps in the cloud-free satellite image record' – here you may want to tweak the sentence and specify that cloud and night impact glacier velocity records derived from optical imagery. This is not an issue for glacier velocities derived from SAR imagery (although those that are derived from SAR have their own nuances as well).

We will clarify here that these issues are specific to optical imagery.

L12: 'dramatic changes' can you quantify this here?

Good suggestion. We will clarify this as order-of-magnitude changes in flow speed.

L63: I believe that custom autoRIFT should be in quotation marks.

We will add the quotation marks.

L72: 'in the Yukon' change to 'in Yukon' ('the' is not necessary)

We will remove "the".

L70/75: Can you provide a sentence here that rationalizes the choice of these five glaciers?

We describe the choice of these five glaciers in the "Test Applications" section and believe that it is more appropriate for it to remain there.

L161: Can you expand on this? How many images were manually filtered out? How user intensive is this process? I expect that it is rather time consuming.

Yes, we will include the fraction of images that were manually filtered out and describe the user's process briefly. With the automatic visualization of the imagery using the "cloud_filtering" script, this takes less than 30 minutes for a site (for a 2-year analysis period).

Figure 2: All the Matlab figures should have colourmap labels.

For the figures under "Coregistration" we will add the colormap labels. All other plots show pixel intensities corresponding to the NIR-band imagery, which we will clarify in the figure caption.

L163/164: Why is a DEM needed? Please add a sentence to describe why this is needed for completeness. How were the use of different DEMs determined? For example, for the Svalbard site, ArcticDEM was available, so why was it not used instead of GMTED2010? Maybe it doesn't matter, but there should be some justification for the DEMs used at each location. How much to the resolution of the DEM matter in terms of the quality of the final velocity maps?

The DEMs are used to calculate glacier surface slopes (dh/dx and dh/dy) to guide the feature-tracking search directions. As long as the DEM captures the relative changes in glacier surface slope, it should be sufficient. The DEM resolution does not matter much. They get resampled to the desired chip sizes (> 100 m) to be compatible with the required autoRIFT input. We will add a sentence clarifying this here.

L269/270: Can you clarify this sentence; you are comparing median stable surface errors from custom autoRIFT to maximum (SSE?) errors from ITS_LIVE? So, are you comparing the same thing?

The ITS_LIVE errors are calculated as the standard error in the component velocities relative to stable surface velocities, rather than using the stable surface velocities themselves. Since we use different stable surface masks and different error metrics, they are not directly comparable and we will retract this comparison from the text.

Figure 6: It feels to me that the scaling for a)b) and c)/d) and e)/f) should be done independent of each other. For example the maximum of c) and d) should be set to 20 m/day and for e) and f) it should be set to (maybe) 15 m/day. I know that you want consistency to compare across these three glaciers, but to me the more important aspect to showcase is the custom autoRIFT performance at each site. At present, with the current scaling applied, the reader cannot see the variations across the glacier in c) and e) because all the variation falls within the purplish colours.

We agree with this suggestion and will adjust Figures 6 and 7 to show the full range of speeds for each site.

**Review 2**

In this paper, the authors present a customized/modified version of NASA autoRIFT, along with the inclusion of PlanetScope (PS) imagery, to produce improved records of glacier flow (and flow instability) for five surge-type glaciers. In nearly all cases, the customized pipeline and additional imagery provided a greatly improved time series of glacier flow speeds, as well as clear improvements to the consistency and structure of the derived speeds. I found the manuscript both well-written and easy to read, and think that it should be published after consideration of minor suggestions/revisions.

l. 7,26: spell out ITS_LIVE at first use (in both abstract and introduction)

We will define the ITS_LIVE acronym (Inter-Mission Time Series of Land Ice Velocity and Elevation) in both initial instances.

l. 28: should the reference here be Gardner et al. (2018), rather than 2023?

The Gardner et al. (2023) reference is for the dataset itself. We can also add the 2018 paper citation associated with it.

l. 48,100, elsewhere: to override the annoying way that LaTeX parses non-standard author names, wrap the name in two curly brackets, e.g. {{Planet Labs}}

Good tip, thank you!

l. 89: how many images does this correspond to? Can you give a range, similar to the file sizes?

The number of images really varies depending on the size of the glacier. Furthermore, the number of tiles versus images differ. There may be 20+ downloadable image tiles for a given day that will end up comprising a single image at the end of the planet_tile2img pipeline (see Fig. 4). Since the answer is complicated, we can provide a broad range and refer the reader to Figure 4 for the nuances mentioned above.

l. 98: What is the purpose of this geotiff - does it need to be, as mentioned later (l. 103-105), a re-interpolated satellite image of the area, or is it enough to just have a "blank" image with the desired resolution and extent? If the latter, would it not just be easier to specify an extent and spatial resolution?

The 5-m geotiff can be blank but needs to have the correct spatial referencing. One could, in theory, create a blank image with the correct extent, resolution, and georeferencing. For simplicity, the user can re-use the DEM and resample it to the desired resolution. We will clarify this in the text.

l. 105: include the version of GRASS that you have used and a citation (e.g., https://zenodo.org/records/10817962)

Ok, we will include the GRASS version and citation here.

l. 130: why would the download step fail based on the number of images?

The failure is due to the connection time out associated with downloading a large number of images through the planet API. We can clarify this in the text.

l. 137-144: this seems to assume that the georeferencing errors do not vary spatially throughout the images. Is this really the case (I suspect not, as it will be related in part to the DEM used for orthorectification), or is it simply "good enough" for most applications to make this assumption?

The coregistration for georeferencing correction is simply there to bolster the alignment of the sections of the glacier where there is not likely to be complicated terrain relative to the surrounding areas. Thus, the targeted portions of the image are less likely to be affected by large orthorectification errors resulting from the DEM. Ultimately, these errors should be a small fraction of the displacements calculated for fast flow (> 5m/d). More detailed corrections could be explored for sites where capturing slow ice flow is important.

l. 144: this is more of a thought/suggestion for future development, rather than for the manuscript. Instead of choosing the second image for co-registration, you could instead have the user provide (or download) a reference image (e.g., Sentinel-2), then co-register each of the PS tiles to that reference image. This way, you aren't left making the assumption that the "first" image or swath in the pair is correctly georeferenced.

This is a clever suggestion, similar to the harmonization strategy to correct Planet radiometry errors but using the Sentinel-2 georeferencing instead.

Figure 2: can you indicate in the caption what the calculated offsets for the coregistration step were?

Yes, we will add the offset to the Figure 2 caption.

Table 1: Do you have an idea/indication of the impact of DEM quality/resolution on the final output? Additionally, you have used both GMTED2010 and GTOPO30, although GMTED2010 is meant to be a replacement/updated for GTOPO30, which is quite coarse in many areas, especially mountainous regions. Is it just the case that the differences between those two datasets for South Rimo are negligible?

That is a reasonable concern, one that the other reviewer was also curious about. The DEMs are used to guide the feature-tracking search directions, so as long as the DEM captures the relative changes in the glacier's surface slope sufficiently, it will improve the CautoRIFT results. The DEMs get resampled to the chipsizes (> 100 m) prior to the autoRIFT input anyways, because autoRIFT uses the DEM spatial grid to produce the velocity map. Thus, the original resolution of the DEMs is not very important. We will make this clearer in the text.

l. 206-210: can you give an indication of the improvement/error reduction that is achieved using the optional coregistration steps?

The error improvement will strongly depend on the glacier size and the surrounding terrain complexity. We have set the maximum georeferencing error for the Planet images as 2 pixels (10 meters), so the maximum displacement improvement is 10 meters over the date separations used of 5 to 60 days. For a glacier speed of 5 m/d, the total displacement would be 25 to 300 meters. 10 meters of georeferencing error reduction would correspond to 3% to 40% of the displacement. We will describe this succinctly in the text.

Figure 5: Can you include an indication of the ITS_LIVE coverage here, perhaps as a line plot alongside the histograms? Or will it be the same as the S2/LS bars, as this may only show the potential pairs rather than the successful pairs?

Rather than complicating this figure, we will add horizontal error bars to Figures 6 and 7 to illustrate the date separation and temporal coverage of the ITS_LIVE data.

l. 252-268: would it be possible to include a table summarizing the statistics derived for each time series for each glacier from the custom autoRIFT results and the ITS_LIVE record? In other words, one row for each glacier and two sets of columns showing the peak speed, average speed during the surge initiation (if possible), median SSE, and number of observations derived from your results and ITS_LIVE based on the points shown in Figures 6 and 7? I think this would help quickly illustrate the clear improvements in coverage that your approach provides.

The suggested table will make a good addition to the manuscript. We will add this in as Table 2.

l. 269-270: this appears to be a comparison of the median errors against the maximum errors - it would be better to compare like against like here.

The ITS_LIVE errors and our stable surface errors are calculated differently. The ITS_LIVE errors are calculated as the standard error in the component velocities relative to stable surface velocities, rather than using the stable surface velocities themselves. Since we use different stable surface masks and different error metrics, they aren't directly comparable. After reflecting upon this concern and the suggestion from the other reviewer, we will retract this direct comparison from the text.

Figures 6,7: I find the improvement in resolution hard to see in the right-hand column - perhaps increasing the size of the markers for the ITS_LIVE observations would help better illustrate where they are absent?

We intend to add horizontal error bars to all the data points to indicate the date interval for each corresponding image pair, which will make the ITS_LIVE observations easier to see. We will also increase the y-scaling of panel of these plos as per the other reviewer's suggestion. These changes should make the differences between ITS_LIVE and CautoRIFT in Fig. 6 and 7 more clear.

l. 293-295: some of these boundaries appear to be between different tiles, rather than swaths - is this a radiometric issue, or a georeferencing/orthorectification issue?

The systematic shift throughout the tile footprint is more indicative of a georeferencing issue. If this is observed to be affecting the velocities over the glacier substantially, the user can try additional coregistration steps, including prior to stitching along-swath.

l. 307-309: can you quantify how much higher the SSE was for the S2 velocity map?

Yes, the SSE for S2 was 0.3 compared to the PS SSE of < 0.1. We will add this to the manuscript text here.

l. 312-315: would it be possible to illustrate the improved feature resolution in a supplemental image that shows the small-scale crevasses in a PS image vs an S2 or Landsat image?

Yes, this would complement the rest of the paper well. We will add it to the supplement.

Figure A3: can you also show the S2 map alongside the PS map here, as a reminder

Sure, that would help reduce the extra white space in the figure too. Thanks for the suggestion.